# *LreEF1A4*, a Translation Elongation Factor from *Lilium regale*, Is Pivotal for Cucumber Mosaic Virus and Tobacco Rattle Virus Infections and Tolerance to Salt and Drought

**DOI:** 10.3390/ijms21062083

**Published:** 2020-03-18

**Authors:** Daoyang Sun, Xiaotong Ji, Yong Jia, Dan Huo, Shiying Si, Lingling Zeng, Yanlong Zhang, Lixin Niu

**Affiliations:** 1College of Landscape Architecture and Arts, Northwest A&F University, Yangling 712100, China; sundaoyang@nwsuaf.edu.cn (D.S.); jixiaotong@nwsuaf.edu.cn (X.J.); huodanstu@nwsuaf.edu.cn (D.H.); sishiying@nwsuaf.edu.cn (S.S.); zll874961289@nwsuaf.edu.cn (L.Z.); 2State Agricultural Biotechnology Centre, School of Veterinary and Life Sciences, Murdoch University, Perth 6150, Australia; y.jia@murdoch.edu.au

**Keywords:** eukaryotic translation elongation factor, *Lilium regale*, petunia, cucumber mosaic virus, tobacco rattle virus, salt, drought

## Abstract

Eukaryotic translation elongation factors are implicated in protein synthesis across different living organisms, but their biological functions in the pathogenesis of cucumber mosaic virus (CMV) and tobacco rattle virus (TRV) infections are poorly understood. Here, we isolated and characterized a cDNA clone, *LreEF1A4,* encoding the alpha subunit of elongation factor 1, from a CMV-elicited suppression subtractive hybridization library of *Lilium regale*. The infection tests using CMV remarkably increased transcript abundance of *LreEF1A4*; however, it also led to inconsistent expression profiles of three other LreEF1A homologs (*LreEF1A1–3*). Protein modelling analysis revealed that the amino acid substitutions among four LreEF1As may not affect their enzymatic functions. *LreEF1A4* was ectopically overexpressed in petunia (*Petunia hybrida*), and transgenic plants exhibited delayed leaf and flower senescence, concomitant with increased transcription of photosynthesis-related genes and reduced expression of senescence-associated genes, respectively. A compromised resistance to CMV and TRV infections was found in transgenic petunia plants overexpressing *LreEF1A4*, whereas its overexpression resulted in an enhanced tolerance to salt and drought stresses. Taken together, our data demonstrate that *LreEF1A4* functions as a positive regulator in viral multiplication and plant adaption to high salinity and dehydration.

## 1. Introduction

Eukaryotic translation elongation factor 1 alpha (eEF1A) is one of the four subunits composing the elongation factor 1, as well as a member of the G protein superfamily [1,2]. It is a conserved cytoplasmic protein widely present in eukaryotic organisms. The abundance of eEF1A in normal growing cells is second only to that of actin, constituting up to 1–3% of the total protein content [3]. As a regulatory enzyme, eEF1A plays a canonical role in the polypeptide chain elongation phase of the protein-synthesis pathway [4]. It catalyzes the delivery of aminoacyl-tRNA to the A-site of the elongating ribosome in a guanosine 5′-triphosphate (GTP)-dependent manner during protein translation. This binding action induces the GTPase activity of eEF1A to cause GTP hydrolysis. Subsequently, the eEF1A-bound guanosine 5′-diphosphate (GDP) is dissociated from the ribosome, and is replaced with the active GTP. The switch between GDP and GTP promotes a further eEF1A-involved enzymatic catalysis for the binding of aminoacyl-tRNA to the acceptor site of the ribosome [5,6].

Aside from the central role in protein synthesis, eEF1A has been revealed to participate in other cellular processes. Direct in vitro interactions of eEF1A with a few proteins, including tubulin [7], calmodulin [8], and actin [9], have been reported. eEF1A may be a potent regulator of microtubules and actin cytoskeleton by binding and bundling filamentous actin (F-actin) [10,11]. The F-actin bundling activity of eEF1A has been shown to be reversibly regulated by a Ca^2+^/calmodulin-sensitive mechanism [12]. Excess eEF1A hampers the F-actin localization and microtubule organization, leading to aberrant cell morphology and further growth defects [13]. Overexpression of *eEF1A* affects the actin cytoskeleton organization and also the cell morphology through the alteration of actin distribution [14]. In addition, eEF1A appears to function as a critical mediator of signal transduction within cells. PIK-A49, a homolog of eEF1A, activates phosphatidylinositol 4-kinase, which is a key enzyme in the growth factor-mediated signal transduction cascade [15]. eEF1A displays high sensitivity to the variation in Ca^2+^ and pH, and probably serves as a crucial downstream target for Ca^2+^ and lipid-involved signal transduction pathway [16]. eEF1A may contribute to the regulation of the DNA replication/repair protein network [17]. A recent study showed that elevated *eEF1A* levels resulted in aberrant spindle formation and chromosomal abnormality through a putative interaction with the dynactin complex [18]. These findings indicate that eEF1A possesses multifunctional properties. Furthermore, much effort has been made to understand the regulatory roles of eEF1A in response to biotic or abiotic stresses.

Some evidence has suggested that plant viruses can recruit host eEF1A to perform their efficient infections due to the limitation on viral genome size [19]. Specific interactions between eEF1A and viral components have been described previously. An earlier research demonstrated that eEF1A interacts with tyrosylated RNA of brome mosaic virus [20]. Since then, eEF1A has been identified as an interactor of viral genomic RNAs or replication proteins of plant viruses, including turnip yellow mosaic virus [21], tobacco mosaic virus (TMV) [22,23], turnip mosaic virus [24], tomato spotted wilt virus (TSWV) [25], and soybean mosaic virus (SMV) [26]. The impact of eEF1A-virus interaction on viral accumulation has also been investigated. For example, the down-regulation of *eEF1A* substantially decreases the TMV levels in *Nicotiana benthamiana*, because of the interaction of eEF1A with 3′-terminal genomic RNA and RNA-dependent RNA polymerase of TMV [27,28]. In the case of the response to abiotic factors, low temperature and heat shock induce the *eEF1A* expression in winter barley [29] and spring wheat [30], respectively. The *AtEF1a* knock-out mutant of Arabidopsis displays higher sensitivity to NaCl stress, and conversely, the plants overexpressing *AtEF1a* are more tolerant to salt [31]. Heterologous expression of a maize plastid *EF*-*Tu*, the counterpart of cytosolic eEF1A, produces fewer symptoms of heat injuries following exposure to high temperature in wheat [4]. These observations reveal an essential participation of eEF1A in plant life cycle to cope with multiple environmental stresses. However, no such eEF1A gene has been reported in ornamental plants.

Lily, an important floricultural plant in the commercial cut-flower market, belongs to the genus *Lilium* of the family Liliaceae. Its economic significance is reflected not only by the aesthetic and aromatic values of the flowers, but also by the nutritional and therapeutic benefits of its fleshy bulb scales [32]. The genus *Lilium* is comprised of more than 80 wild species [33], among which 55 have been confirmed to be distributed in China [34]. A rare species endemic to China, *Lilium regale* Wilson, is considered a precious germplasm for lily breeding due to its outstanding performance in defense against viruses, fungi, and abiotic stresses [35]. It has been indicated that *L. regale* exhibits strong resistance to cucumber mosaic virus (CMV, genus *Cucumovirus*), one of the most prevalent viruses in lilies [36]. It is also supported by our recent finding that *L. regale* has much lower incidence of viral disease than the other nine species tested under natural infection [37]. *L. regale* is highly resistant to *Fusarium oxysporum* f.sp. *lilii*, a soil-borne fungal pathogen causing root or bulb scale rot [38]. Regarding its response to abiotic stimuli, Zhao et al. [39] reported *L. regale*’s excellent tolerance to salinity and drought. To date, few studies have focused on the dissection of defensive mechanisms against these adverse stresses in *L. regale*.

In previous work, we employed the suppression subtractive hybridization (SSH) approach to identify a cluster of genes that are significantly up-regulated in CMV-infected *L. regale* leaves. We have used petunia as a heterologous model system, owing to the recalcitrance in genetic transformation of lilies, to characterize the functions of these candidate genes. Of them, a general control non-derepressible (GCN)-type adenosine triphosphate (ATP)-binding cassette transporter gene, *LrABCF1*, has proven to be a crucial regulator of defense responses against CMV, tobacco rattle virus (TRV), and *Botrytis cinerea* [37]. In this study, we report the important involvement of another gene encoding a putative eEF1A in CMV and TRV infections. A positive role of this gene in salt and drought tolerance is also described.

## 2. Results

### 2.1. Isolation of Full-Length LreEF1A4 cDNA

A 725-bp EST sequence, encoding a putative translation elongation factor 1 alpha (eEF1A), was identified among up-regulated genes in CMV-infected leaves of *L. regale* from an SSH analysis [37]. RACE-based amplification was performed to obtain its full-length cDNA sequence, designated *LreEF1A4* (GenBank accession no. MT083900, Appendix A). The *LreEF1A4* cDNA includes a complete open reading frame encoding 447 amino acids. Sequence analysis demonstrated that LreEF1A4 possesses four regions (G-1 to G-4) implicated in GDP/GTP exchange and GTP hydrolysis, three elements in the GTP-binding domain, and one GTP-binding elongation factor signature motif (Figure 1A). Phylogenetic tree revealed that LreEF1A4 was highly similar to three copies (LreEF1A1–3) from *L. regale*, and homologs from *Arabidopsis thaliana* (AteEF1A1–4), *Oryza sativa* (OseEF1A1–4), *Gossypium hirsutum* (GheEF1A1–9), *Zea mays* (ZmeEF1A1–7), and other species (Figure 1B).

### 2.2. Protein Structure Analysis of LreEF1A4

Sequence alignment of the four LreEF1A copies (LreEF1A1–4) in lily identified a total of 24 amino acid sites displaying residue substitutions (Figure 2A). To investigate whether the amino acid substitutions affect the enzymatic functions of different LreEF1A homologs, a high-quality structural model of LreEF1A4 was built using the mammalian elongation factor structure (Protein Data Bank: 5LZS) as the template. The 5LZS structure, consisting of three major domains (Figure 2B–D), was chosen as the template due to it being in complex with the enzyme substrate GDP and the other interacting components in the ribosomal machinery. The spatial locations of these substitution sites were examined in this generated LreEF1A4 model. As shown in Figure 2A,B, most of these amino acid substitutions (16/24) were found located in the loop regions. No amino acid substitution site in the substrate-binding (GDP) domain I displayed any interacting potential with the superimposed substrate (Figure 2C). All of the identified amino acid changes were located on a single side of the protein model (Figure 2D), facing away from the ribosomal interacting side (Figure 2E). These results indicate that the amino acid changes may not affect the interaction of different LreEF1As with the other components in the ribosomal complex.

### 2.3. CMV Infection, Abiotic Stresses, and Hormone Treatments Change LreEF1As Expression

To compare the expression pattern difference of *LreEF1A4* with three other copies, the healthy leaves of *L. regale* at the two-leaf stage were inoculated with CMV and treated with various abiotic stressors and plant growth regulators. *LreEF1A4* was rapidly and dramatically up-regulated upon CMV infection, whereas a significant down-regulation of *LreEF1A2* transcripts was observed. In contrast, expression levels of *LreEF1A1* and *LreEF1A3* remained stable during the entire infection period (Figure 3A). Transcript abundances of all four LreEF1A genes increased following treatments with high salinity, dehydration, and low temperature, with *LreEF1A4*, *LreEF1A1*, and *LreEF1A2* reaching relatively higher levels at 12 or 24 h post treatment (HPT), respectively (Figure 3B–D). A remarkable decrease in the expression of four *LreEF1A*s occurred under heat stress (Figure 3E). For treatments with stress-associated hormones, *LreEF1A1–4* were all activated following exposure to ethylene (Eth) and this activation was maintained thereafter until 24 HPT (Figure 3F). Transcripts of *LreEF1A1–4* were also induced by gibberellic acid (GA_3_) treatment, with a peak at 6 HPT and a subsequent decline (Figure 3H). No significant alteration of transcription was found for *LreEF1A*s in response to salicylic acid (SA) and abscisic acid (ABA) (Figure 3G,I).

### 2.4. Ectopic Overexpression of LreEF1A4 Delays Leaf and Flower Senescence

To study the function of *LreEF1A4*, a genetic transformation assay was performed in petunia, a model testing platform for molecular biology research of ornamental crops. We generated three transgenic petunia lines (1B, 2A, and 8E), in which the constitutive overexpression of *LreEF1A4* was confirmed through semi-quantitative reverse transcription-polymerase chain reaction (RT-PCR) and quantitative real-time PCR (Figure 4A,B). The bottom leaves of wild-type (WT) petunia plants displayed severe yellowing phenotypes, while the leaves of *LreEF1A4*-overexpressing transgenic lines at the identical developmental phase still remained green (Figure 4C). Total chlorophyll content in the leaves of transgenic lines was much higher than that in the leaves of WT lines (Figure 4D). An increase in transcript levels of photosynthesis-related genes, *ribulose-1,5-bisphosphate carboxylase*/*oxygenase large subunit* (*rbcL*) and *photosystem I chlorophyll a apoprotein* (*psaA*), was found in transgenic lines (Appendix A). *LreEF1A4* overexpression remarkably extended the longevity of attached unpollinated flowers of petunia compared to WT control (Figure 4E,F), and reduced the expression of senescence-associated genes, *SAG12* and *SAG29*, in the corollas (Appendix A).

### 2.5. LreEF1A4 Affects Resistance to CMV

Given the pronounced up-regulation of *LreEF1A4* in *L. regale* plants upon CMV infection, the role of *LreEF1A4* in defense response against CMV was then examined. CMV strain was used to inoculate WT and transgenic petunia plants with *LreEF1A4* overexpression. At 14 days post inoculation (DPI), the overexpression lines exhibited more distortion of uppermost systemically-infected leaves and stunting of affected plants than WT lines (Figure 5A). An increased number and larger size of necrotic lesions were found in the infected leaves of transgenic plants (Figure 5B,C). This symptom variation was in accordance with the measured accumulation of CMV coat protein (*CMV-CP*) transcripts, which were significantly increased in *LreEF1A4*-overexpressing lines at 10 and 14 DPI, compared to WT control (Figure 5D).

To validate the involvement of *LreEF1A4* in CMV proliferation, we performed virus-induced gene silencing (VIGS) assay using tobacco rattle virus (TRV) as a vector to down-regulate *PheEF1A*, the petunia orthologue of *LreEF1A4*. After CMV inoculation, the TRV-*PheEF1A*-infiltrated petunia plants had much milder symptoms, as demonstrated by less leaf mottling and alleviated plant stunting, than the plants infiltrated with TRV empty vector (EV) (Appendix A). Quantitative real-time PCR analysis verified the reduced abundances of *PheEF1A* and *CMV*-*CP* transcripts in TRV-*PheEF1A*-infected petunia plants upon CMV inoculation (Appendix A).

### 2.6. LreEF1A4 Affects Resistance to TRV

To determine whether *LreEF1A4* has a similar influence on the multiplication of other virus strains, we inoculated the transgenic and control lines with a TRV vector expressing green fluorescent protein (TRV-GFP). GFP can act as a reporter to visualize the TRV accumulation. Under excitation using a blue light-emitting flashlight, the inoculated leaves of transgenic plants overexpressing *LreEF1A4* showed more fluorescent foci than those of WT plants at 4 DPI (Figure 6A). The fluorescent area was visibly larger in transgenic lines than that in the WT lines (Figure 6B). In line with the fluorescence, significantly higher accumulation levels of TRV RNA1 and RNA2 were found in the overexpression lines than in the control lines (Figure 6C). Meanwhile, decreased TRV accumulation was detected in *PheEF1A*-silenced petunia plants infiltrated with TRV-*PheEF1A* compared to the control plants infiltrated with TRV-EV (Appendix A).

### 2.7. LreEF1A4 Participates in the Tolerance to Salt and Drought

To further dissect the role of *LreEF1A4* in abiotic stress tolerance, we treated the seeds or seedlings of WT and *LreEF1A4*-overexpressing transgenic petunia lines with NaCl, dehydration, and low temperature. At 10 and 20 days after sowing (DAS), over 90% of WT and transgenic seeds germinated and maintained continuous growth on the control Murashige and Skoog (MS) plates without NaCl, with no significant difference in germination rates between WT and transgenic lines (Figure 7A,B). In the NaCl-supplemented medium, the high salinity pressure strongly triggered abnormal growth and suppressed the germination of WT seeds in contrast with transgenic ones (Figure 7A). Approximately 12% of WT seeds germinated under salt stress, which was drastically lower than the germination rates of petunia lines overexpressing *LreEF1A4* (Figure 7B). After the drought treatment, transgenic petunia lines exhibited much milder leaf wilting and higher total fresh weight than WT lines (Figure 7C,D). No significant difference in the symptoms and electrolyte leakage was found between WT and transgenic plants when exposed to cold stress (Appendix A).

## 3. Discussion

Large-scale transcriptome analysis allows the identification of candidate genes differentially expressed in plants under various stress conditions. Considering the outstanding resistance characteristics of *L. regale*, a number of studies have been carried out to fully explore the resistance-related candidate genes from this wild *Lilium* species by the researchers. Vast amount of differentially-expressed transcripts related to soil-born fungus *F. oxysporum* [40] or phytopathogenic fungus *Botrytis elliptica* [41] in *L. regale* have been excavated through the SSH or RNA-seq approach. We originally performed the CMV-induced SSH analysis for the identification of key determinants of viral defense in *L. regale* [37]. As a result of our screening, one up-regulated transcript, *LreEF1A4*, in *L. regale* during CMV infection was isolated and functionally characterized. The data obtained from our work support that this gene plays essential roles in viral infection and abiotic stress tolerance (Figure 5, Figure 6 and Figure 7 and Appendix A).

It is well recognized that the eEF1A family comprises multiple copies in plants. Arabidopsis [42] and rice [43] contain four eEF1A homologs, while there are two in soybean [44] and carrot [45]. Four to eight copies of eEF1A have been described in tomato [46], and nine have been reported in cotton [47]. The most documented number of eEF1A homologs is 10–15 in maize [48]. Animals and fungi also exhibit a variable amount of eEF1A genes [2]. The polygenic feature of eEF1A family is presumably crucial to guarantee the efficacy and fidelity of protein synthesis under various conditions [47]. Apart from SSH-derived *LreEF1A4*, we identified three other distinct genes, *LreEF1A1–3*, belonging to the eEF1A family. Due to a gigantic genome size and limited genomic information in *Lilium*, more eEF1A genes may be identified in this species.

The amino acid sequences of eEF1As are highly conserved across different living organisms (Figure 2A). Despite a number of amino acid substitutions among the four lily LreEF1A homologs, most of them were distributed in the loop regions based on the LreEF1A4 structural model we built. The loops correspond to the most flexible regions in a protein structure, and are under much less structural constraints. Compared to the mammalian counterparts, another notable finding is a 12-amino acid (aa) deletion occurring in LreEF1A proteins (Figure 2A). This deletion is conserved in the Arabidopsis eEF1A as well, and may represent a unique feature for plant elongation factors. The deleted 12-aa fragment is also located within a loop region in the domain I of mammalian eEF1A, and is exposed to the solvent surface opposing to the ribosomal binding side (Figure 2E). Therefore, it is less likely to cause significant functional alteration in plant eEF1As. Collectively, protein modelling analysis indicates that the amino acid variation among four LreEF1As seems not to change their enzyme function. This suggestion is consistent with the strictly conserved functions of the elongation factor protein family in protein synthesis [49]. However, the functional assessment of diverse LreEF1As in defense responses merely by means of protein structural comparison is not completely convincing. The specific functional divergence between members of a gene family may be affected by other factors, such as the expression state [50].

It is noteworthy that the transcriptional responses of four *LreEF1A*s to CMV infection varied in *L. regale* (Figure 3A). Thus, we hypothesize that expression patterns of these LreEF1A homologs perhaps contribute to their different functions in response to CMV. Currently, few attempts have been made to accurately unravel the functional discrepancy among eEF1A copies, especially in plant defense against viral pathogens. Moreover, we found almost uniform expression profiles of *LreEF1A1–4* in *L. regale* under several types of abiotic stresses (salt, drought, cold, and heat) and stress-related hormone treatments (Figure 3B–I), implying the functional similarity of various *LreEF1A*s in abiotic stress responses. Elevated abundances of *LreEF1A1–4* transcripts following exposure to salt, drought, and cold were in line with the observations in yeast [51], sheepgrass [52], maize [53], or rice [54]. These data suggest the possible involvement of *LreEF1A*s in tolerance to these stresses. The reduced hypersensitivity to high concentrations of NaCl and drought stresses conferred by *LreEF1A4* overexpression in our work supports this notion (Figure 7). These findings are in agreement with the previous reports on the positive involvement of *AtEF1a* in the tolerance to high salinity in yeast and Arabidopsis [31], or the positive role of *GmEF4* encoding eEF1A in the salt and drought tolerance in soybean [55]. However, Rausell et al. [56] reported that *BveEF1A* transcripts from the halotolerant plant sugar beet were not up-regulated under salt stress. Our observation of the diminished *LreEF1A*s transcripts by the heat shock also differs from the previous results that high temperature activates the expression of rice *eEF1A* [54]. The inconsistent stress treatment design when comparing our experiments with others could probably explain such expression variability.

Many pieces of evidence have revealed a close relationship between virus infection and host eEF1As, which mostly assist replication of RNA viruses by interacting with viral components. For instance, the eEF1A inhibitor reduces the RNA synthesis activity of TSWV, supporting a potential interaction between eEF1A and this viral RNA [25]. The P3 protein of SMV interacts with eEF1A to boost the unfolded protein response and viral virulence, and *eEF1A* silencing suppresses the spread of SMV in soybean [26]. In animals, eEF1A appears to have a similar function in viral pathogenesis. eEF1A plays an indispensable role in the late steps of human immunodeficiency virus type-1 reverse transcription by interacting with viral reverse transcriptase and integrase [57]. eEF1A1 interacts with NS3 and NS5 proteins of Japanese encephalitis virus to stabilize the components of the viral replicase complex and promoting viral replication [58]. When reviewing the previous literatures, the controversy concerning the effect of eEF1As on virus propagation still exists. Silencing of *eEF1A* via TRV-based VIGS provokes the collapse of tomato resistance to tomato yellow leaf curl virus [59]. Up-regulation of *eEF1A* decreases the replication of classical swine fever virus (CSFV) with an interaction with the NS5A protein, whereas the knockdown of *eEF1A* elicits an enhanced effect on CSFV accumulation [60]. The mechanisms underlying these apparently conflicting findings are not yet well understood.

To date, eEF1As have been reported to positively affect the propagation of multiple virus types. Nevertheless, little is known about the influence of eEF1As on CMV and TRV infections, both of which are highly prevalent pathogens with wide host ranges in plants. Our work of genetic transformation and TRV-based VIGS assays verified the enhanced susceptibility to CMV and TRV in *LreEF1A4*-overexpressing transgenic petunia plants, and at the same time the reduced susceptibility to the two viruses in its orthologous gene *PheEF1A*-silenced petunia plants (Figure 5, Figure 6, and Appendix A). The unanswered question here is whether LreEF1A4 interacts with the genomic RNAs or proteins of CMV and TRV. Future work will include an investigation of which component of these two viruses targets LreEF1A4.

It should be mentioned that *LreEF1A4* overexpression led to delayed leaf aging and petal wilting (Figure 4C,E). Increased transcript abundances of a few photosynthesis-related genes in leaves and decreased expression of senescence-associated genes in floral tissues of transgenic lines corroborate these findings (Appendix A). Thus, we conclude that *LreEF1A4* may function in the regulation of leaf and flower senescence. Senescence is the terminal phase in the development of a biological structure, normally synonymized with the programmed death of all living cells [61]. eEF1A has been identified as a vital regulator of programmed cell death (PCD) in addition to its canonical role in translation [62]. The slowed onset of chlorophyll degradation-induced leaf yellowing prior to PCD in overexpression lines provides compelling evidence on *LreEF1A4′*s positive role in controlling leaf senescence. Although leaves and flowers share common evolutionary origins, the regulatory mechanism underlying their senescence may be largely different in the aspects of reactive oxygen species, nutrient remobilization, inductive factors, or phytohormones [63]. We noted that exogenous application of Eth and GA_3_ induced the expression of *LreEF1A4* (Figure 3F,H). There has been discussion on the crosstalk between Eth and GA_3_ during corolla senescence in ethylene-sensitive petunias [64]. The functional significance of LreEF1A4 in modulation of flower senescence may be attributed to the interplay between LreEF1A4 and these hormones. More work will be required to clarify whether *LreEF1A4* affects the endogenous hormone secretion in senescencing plant organs.

## 4. Materials and Methods

### 4.1. Plant Materials and Growth Conditions

*L. regale* seeds, harvested from virus-free plants grown in lily germplasm resources nursery of Northwest A&F University (Yangling, China), were first soaked in warm water at 30 °C for 48 h to improve germination efficiency. Then, they were planted in a 36-well plastic tray filled with sterile soil mix (peatmoss:perlite:vermiculite = 2:1:1, by vol.). *P. hybrida* cv. ‘Mitchell Diploid’ seeds, purchased from Goldsmith Seeds Inc. (Gilroy, CA, USA), were directly sown in the same plastic tray containing soil mix. The seeds were germinated in a growth chamber at 25/22 °C day/night temperature with a 60% relative humidity and light/dark cycle of 16/8 h. Two-leaf-stage lily seedlings were inoculated with viruses or treated with abiotic factors and hormones for assessment of *LreEF1A*s expression patterns. Young petunia leaves 3–8 from terminal were used for ectopic overexpression experiment. Four-leaf-stage WT and transgenic petunia plants were subjected to CMV, TRV-GFP, or TRV-derivative inoculation. Petunia seeds were used for the salt tolerance test, and five-week-old petunia seedlings were used for the examination of drought and cold tolerance.

### 4.2. Viral Inoculation

CMV strains were isolated from the naturally-infected leaves of Oriental hybrid lily cultivar ‘Sorbonne,’ grown in our lily germplasm resources nursery. The infectious inoculum was prepared following a previous description [65]. The viral preparations were mechanically inoculated onto the second newly-emerged healthy *L. regale* leaves, dusted with 400-mesh carborundum powder [66]. After inoculation, the leaves were immediately washed with sterile deionized water to remove excess abrasives. For functional verification of *LreEF1A4* in virus resistance, the CMV inoculum was rub-inoculated onto the fully-expanded young leaves of WT and transgenic petunia plants. TRV-GFP constructs were agro-infiltrated into petunia leaves based on a VIGS assay as characterized below. Transcript levels of CMV coat protein-encoding gene *CMV-CP* and TRV RNA1/RNA2 were determined by quantitative real-time PCR. GFP imaging was performed using a portable blue light-emitting diode flashlight as an excitation source at 450 nm (LUYOR-3260RB, LUYOR, Shanghai, China). Photographs were taken using a Canon EOS 40D digital camera with a bypass emission filter (LUV-495, LUYOR, Shanghai, China). The lesions and fluorescence signals were quantified by measuring pixel sizes of the necrotic and green spots using Photoshop CS6 (Adobe Systems, San Jose, CA, USA) to calculate the relative area.

### 4.3. Abiotic Stress and Hormone Treatments

To examine the expression profiles of *LreEF1A1–4* in response to abiotic stresses and stress-associated hormones, two-leaf-stage *L. regale* seedlings were used for the subsequent treatments. When treated with the salinity and dehydration, the seedlings were irrigated with 200 mM NaCl or 20% PEG6000 solutions at room temperature. For the cold and heat treatments, the plants were placed into a constant-temperature room at 4 and 37 °C, respectively. For the Eth treatment, 8 μL·L^–1^ Eth gas was continuously supplied into a sealed transparent glass box, where the plants were maintained. For the treatments with other hormones, the solutions containing 100 μM SA, 40 μM GA_3_, and 40 μM ABA were sprayed onto the leaves. The second broad true leaves of *L. regale* seedlings were collected at intervals (0, 3, 6, 12, and 24 h) post-treatment for expression analysis. To test the impact of *LreEF1A4* overexpression on salt tolerance, the surface-sterilized seeds of WT and *T_2_* transgenic petunia lines were planted on MS plates containing 200 mM NaCl, and the plates without NaCl were used as the control. Germination rate was measured at a single time point after sowing. For the drought tolerance test, five-week-old plantlets of WT and transgenic petunia lines were grown without watering, and total fresh weight of the whole plant was subsequently measured. For the examination of cold tolerance, petunia seedlings were acclimated at 4 °C for 48 h, and then subjected to −2 °C for 24 h. Electrolyte leakage was determined by measuring the conductivity of treated petunia leaves as previously described [67]. The relative electrolyte leakage was expressed as a ratio of the conductivity before autoclaving to that after autoclaving.

### 4.4. Full-Length Cloning of Four LreEF1As

Based on a cDNA fragment of *eEF1A* from CMV-induced *L. regale* SSH library [37], the gene-specific primers (GSP) and nested gene-specific primers (NGSP) were designed to implement the 3′ and 5′ RACE using a BD SMARTTM RACE cDNA Amplification Kit (Clontech, Mountain View, CA, USA). GSP1 (5′-CGATCTGGAAAGGAACTTGAGAA-3′) and NGSP1 (5′-GAAGAATG GTGATGCTGGCATGA-3′) were used for 3′ RACE. GSP2 (5′-ACATTCTTGACATTAAAGCCAAC -3′) and NGSP2 (5′-CAGTAGGACCAAAGGTAACAACC-3′) were used for 5′ RACE. The resulting sequence harboring complete 3′ and 5′ ends was finally assembled and annotated as *LreEF1A4*. The full lengths of *LreEF1A1*, *LreEF1A2*, and *LreEF1A3* were PCR-amplified using primers (Appendix A) according to the corresponding complete cDNA sequences of three *eEF1A*s (accession no. AF121261, AF181491, and AF181492) from *L. longiflorum*, which were previously deposited in the NCBI GenBank (http://blast.ncbi.nlm.nih.gov/).

### 4.5. Sequence Analysis

The open reading frame regions of four *LreEF1A*s were translated to amino acids using ExPASy (http://web.expasy.org/translate/). Homologous proteins of LreEF1A1–4 were searched using the basic local alignment search tool (BLAST) against the non-redundant GenBank protein databases. A phylogenetic tree analysis was carried out using ClustalW (https://www.genome.jp/tools-bin/clustalw) and MEGA (version 4.0.2, https://www.megasoftware.net/). Alignment of amino acid sequence was conducted using the Multiple Sequence Alignment tool (http://multalin.toulouse.inra.fr/) and further annotated using the ESPript 3.0 tool (http://espript.ibcp.fr/).

### 4.6. Protein Structural Modelling

The protein structural model of LreEF1A4 was generated by homology modelling using SWISS-MODEL (https://swissmodel.expasy.org/). The full-length amino acid sequence of LreEF1A4 was used to search for the structural templates. The top query hit (Protein Data Bank: 5LZS) with a strong amino acid similarity (76.3%) was selected for model building. The obtained protein model was assessed by the Global Model Quality Estimate (GMQE = 0.89) score and Qualitative Model Energy ANalysis (QMEAN Z-score = −1.81) [68], which indicate that the model is of high quality. The final model was validated by Ramachandran plot analysis using PROCHECK (http://www.ebi.ac.uk/thornton-srv/software/PROCHECK). Protein structural superimposition and molecular visualizations were performed using PyMOL (version 1.3r1, Schrödinger, New York, NY, USA).

### 4.7. Semi-Quantitative RT-PCR and Quantitative Real-Time PCR

Extraction of total RNA was performed on lily and petunia leaves through an RNeasy Plant Mini Kit (Qiagen, Valencia, CA, USA). The isolated RNA was purified using RNase-free DNase I (Promega, Madison, WI, USA) for genomic DNA elimination. A NanoDrop ND-2000 spectrophotometer (NanoDrop Technologies, Wilmington, DE, USA) was used to evaluate the quality and concentration of RNA. First-strand cDNA was synthesized from 2–5 μg of RNA samples using PrimeScript RT reagent (Takara, Otsu, Shiga, Japan) with oligo(dT) primer. Semi-quantitative RT-PCR was conducted using Premix Taq DNA polymerase (TaKaRa, Otsu, Shiga, Japan). PCR products were checked by electrophoresis on 1.5% agarose gel stained with GelRed (Biotium, Hayward, CA, USA), and visualized in a Gel Doc XR+ system (Bio-Rad, Hercules, CA, USA) under UV illumination.

Quantitative real-time PCR was performed using the synergy brands (SYBR) Green reagent in a LightCycler480 Real-Time PCR System (Roche Diagnostic, Basel, Switzerland). Gene expression levels were standardized to *Glyceraldehyde-3-phosphate dehydrogenase* (*LrGAPDH*) and *26S ribosomal RNA* (*26S rRNA*) for lily and petunia, respectively. Relative expression was determined through a previously described method [69]. A set of specific primers for expression analysis are listed in Appendix A. PCR primers to sequences in the untranslated regions were used for examination of *LreEF1A*s transcripts, due to small nucleotide variations in their coding regions.

### 4.8. Plasmid Construction

To generate the overexpression construct, a 1395-bp fragment containing the coding sequence of *LreEF1A4* was PCR-amplified using the forward and reverse primers with the *Kpn*I and *Sal*I restriction sites (Appendix A), respectively. The resulting product was inserted into the simple vector pMD19-T and subsequently transferred to a modified plant expression vector pCAMBIA1300 in the sense orientation under the control of CaMV 35S promoter. For the VIGS construct, a 314-bp sequence of *PheEF1A*, the petunia homolog of *LreEF1A4*, was amplified using primers bearing no restriction sites (Appendix A). The amplified fragment was cloned into the silencing vector TRV between *Sac*I and *Xho*I sites through another intermediate vector pUCm-T. The plasmid constructs were sequenced to verify their authenticity.

### 4.9. Plant Transformation

The recombinant *35S::LreEF1A4* plasmid was introduced into *Agrobacterium tumefaciens* LBA4404 by electroporation. The positive colonies were obtained on solid LB plates containing 50 mg·L^−1^ kanamycin after 72 h of bacterial growth at 28 °C. One colony was selected for 48 h of culturing in liquid yeast extract peptone (YEP) media containing 10 g·L^−1^ yeast, 10 g·L^−1^ peptone, 5 g·L^−1^ NaCl, and appropriate antibiotics. Prior to genetic transformation, the agrobacterial concentration was diluted to an OD600 of 0.3–0.4 using liquid LB media without antibiotics. Next, young leaves of petunia cv. ‘Mitchell Diploid’ were transformed and regenerated using the leaf disc method [70]. The transformed plants were screened on MS plates supplemented with 150 mg·L^−1^ hygromycin B. A continuous cultivation was performed until the generation of homozygous *T_2_* lines as previously described [71]. The abundance of transgene transcripts was determined via semi-quantitative and quantitative methods.

### 4.10. Measurement of Chlorophyll Levels

Total chlorophyll content was tested according to the method described previously [72]. Approximately 0.2 g of fresh petunia leaf tissues were ground into powder with liquid nitrogen, and then loaded into a 15-mL centrifuge tube. Chlorophyll was extracted with 10 mL of acetone:anhydrous ethanol (1:1, by vol.) mixture solution in the dark at room temperature for 24 h. Chlorophyll a and b levels were determined based on the absorbance data at 663 and 645 nm, respectively, using a Beckman DU-730 UV-visible spectrophotometer (Beckman Instruments, Palo Alto, CA, USA).

### 4.11. VIGS Assay

The TRV-*PheEF1A* plasmid was electrotransformed into *A. tumefaciens* GV3101. The transformed *Agrobacteria* with kanamycin resistance were cultured in LB media containing 10 mM MES, and 20 μM acetosyringone, 20 mg·L^–1^ gentamicin, and 40 mg·L^–1^ kanamycin at 28 °C for 48 h. The cultures were centrifuged at 4000 rpm for 20 min, and resuspended in the infiltration buffer containing 10 mM MES, 10 mM MgCl_2_, and 200 μM acetosyringone through a gentle shaking for 3 h. The bacterial suspensions bearing the TRV1 and TRV2 constructs were mixed in a 1:1 ratio, and used to inoculate the abaxial leaf surface of WT petunia seedlings through a needleless syringe [73,74,75]. The primers to sequence beyond not only the inserted fragment but also the coding region were used to examine transcript abundance of *PheEF1A* (Appendix A).

## 5. Conclusions

In this study, the SSH analysis of CMV-infected *L. regale* led to the identification of a translation elongation factor *LreEF1A4*. Special attention was given to the role of *LreEF1A4* in responses to viral attack and abiotic stresses. *LreEF1A4* was found to be up-regulated exclusively upon CMV infection compared to three other lily homologs. The transcription of *LreEF1A4* was changeable under treatments of different abiotic stressors and hormones. We also found that ectopic overexpression of *LreEF1A4* resulted in delayed organ senescence and enhanced salt and drought tolerance in petunia. More importantly, we proved that *LreEF1A4* serves as a pivotal participant in CMV and TRV proliferation. Altogether, the data presented here not only shed new lights on the interaction between viral pathogens and host factors, but also provide a valuable reverse genetics strategy for improving antiviral potency of *Lilium* crops, which would ultimately benefit the cut-flower industry.

## Figures and Tables

**Figure 1 ijms-21-02083-f001:**
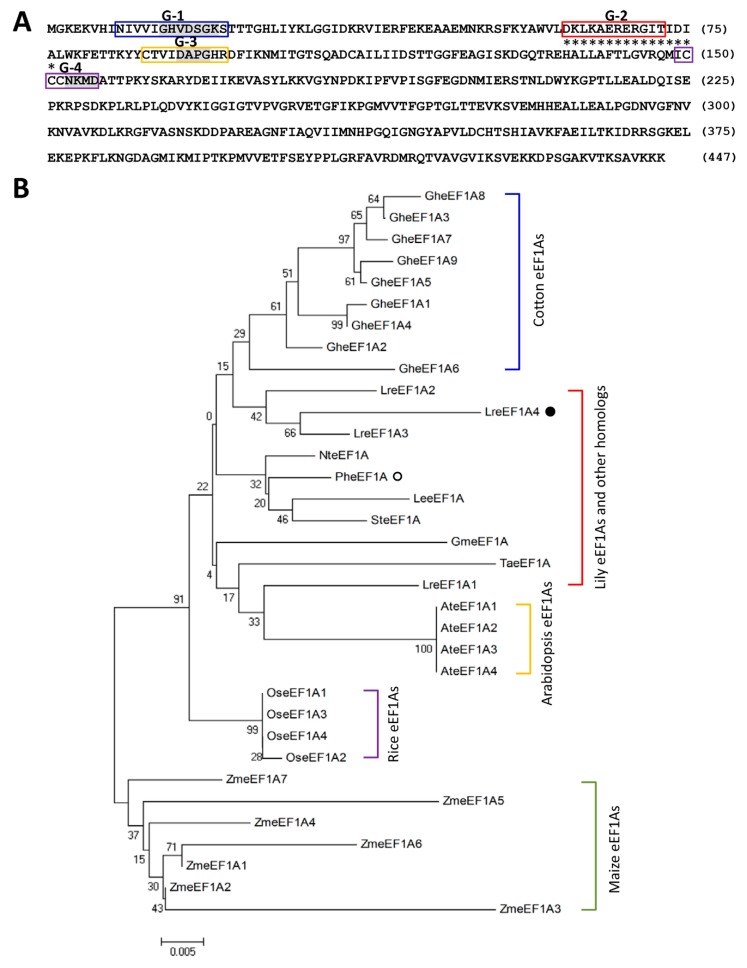
Amino acid and phylogenetic analysis of LreEF1A4. (**A**) Deduced amino acid sequence of LreEF1A4 from *Lilium regale*. The central regions (G-1 to G-4) involved in guanosine 5′-diphosphate/ guanosine 5′-triphosphate (GDP/GTP) exchange and GTP hydrolysis are boxed in blue, red, orange, and purple, respectively. The conserved three elements GXXXXGK (G14–K20), DXPG (D91–G94), and NKXD (N153–D156) in the GTP-binding domain are shaded in grey. Asterisks indicate the GTP-binding elongation factor signature. (**B**) Phylogenetic tree of LreEF1A4 with three other *L. regale* LreEF1As, *Petunia hybrida* PheEF1A (Peaxi162Scf00351g00321, Sol Genomics Network database), *Lycopersicum esculentum* LeeEF1A (P17786), *Solanum tuberosum* SteEF1A (XP_006340223), *Nicotiana tabacum* NteEF1A (XP_016492900), *Glycine max* GmeEF1A (P25698), *Triticum aestivum* TaeEF1A (AQU14666), Arabidopsis AteEF1A1 (At1g07920), AteEF1A2 (At1g07930), AteEF1A3 (At1g07940), and AteEF1A4 (At5g60390), *Oryza sativa* OseEF1A1 (BAA23657), OseEF1A2 (BAA23658), OseEF1A3 (BAA23659), and OseEF1A4 (BAA23660), *Gossypium hirsutum* GheEF1A1 (ABA12217), GheEF1A2 (ABA12218), GheEF1A3 (ABA12219), GheEF1A4 (ABA12220), GheEF1A5 (ABA12221), GheEF1A6 (ABA12222), GheEF1A7 (ABA12223), GheEF1A8 (ABA12224), and GheEF1A9 (ABA12225), and *Zea mays* ZmeEF1A1 (AAF42976), ZmeEF1A2 (AAF42977), ZmeEF1A3 (AAF42978), ZmeEF1A4 (AAF42979), ZmeEF1A5 (AAF42980), ZmeEF1A6 (AAF42981), and ZmeEF1A7 (AAF42982). LreEF1A4 and PheEF1A are highlighted by solid and hollow circles. Boot-strap values were determined as a percentage of 1000 replicates and shown at corresponding branch nodes. Scale bar denotes the number of amino acid substitutions per site.

**Figure 2 ijms-21-02083-f002:**
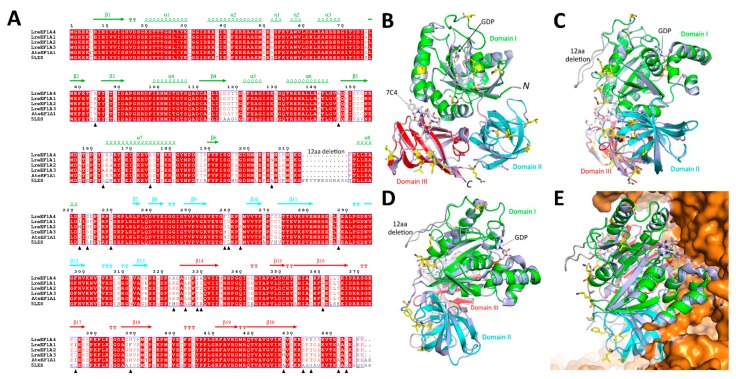
Protein structural modelling of LreEF1A4. (**A**) Amino acid sequence alignment of LreEF1A4 with other homologs, including *Lilium regale* LreEF1A1–3, Arabidopsis AteEF1A1, and mammal 5LZS. The secondary structure is displayed above the sequences in green (domain I), cyan (domain II), and red (domain III). Amino acid substitution sites within LreEF1A1–4 are indicated with solid triangle. (**B–D**) Structural superimposition of LreEF1A4 model with mammalian eEFA1 5LZS (blue) in different angles. (**E**) Spatial orientation of LreEF1A4 in complex with ribosomal proteins (orange). Domain I, II, and III are highlighted in green, cyan, and red, respectively. Amino acid substitution sites within LreEF1A1–4 are shown in yellow sticks.

**Figure 3 ijms-21-02083-f003:**
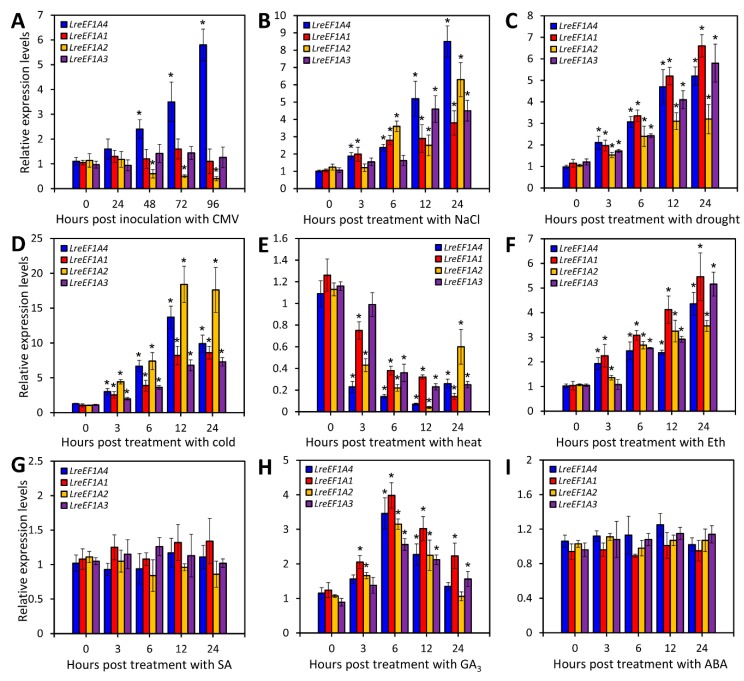
Expression of *LreEF1A*s in *Lilium regale* leaves under cucumber mosaic virus (CMV) inoculation, abiotic stresses, and hormone treatments. Quantitative real-time polymerase chain reaction (PCR) analysis of *LreEF1A1–4* transcript abundances in response to CMV (**A**), 200 mM NaCl (**B**), dehydration (**C**), low temperature (4 °C, **D**), high temperature (37 °C, **E**), 8 μL·L^–1^ ethylene (Eth, **F**), 100 μM salicylic acid (SA, **G**), 40 μM gibberellic acid (GA_3_, **H**), and 40 μM abscisic acid (ABA, **I**) at indicated time points. *L. regale* plantlets at the two-leaf stage were infected with CMV, or treated with various abiotic stressors and plant growth regulators. Transcript levels were standardized to *LrGAPDH*. Error bars represent standard error (SE) of the mean from three independent biological replicates. Asterisks indicate statistical significance as determined by Student’s *t*-test at *p* < 0.05.

**Figure 4 ijms-21-02083-f004:**
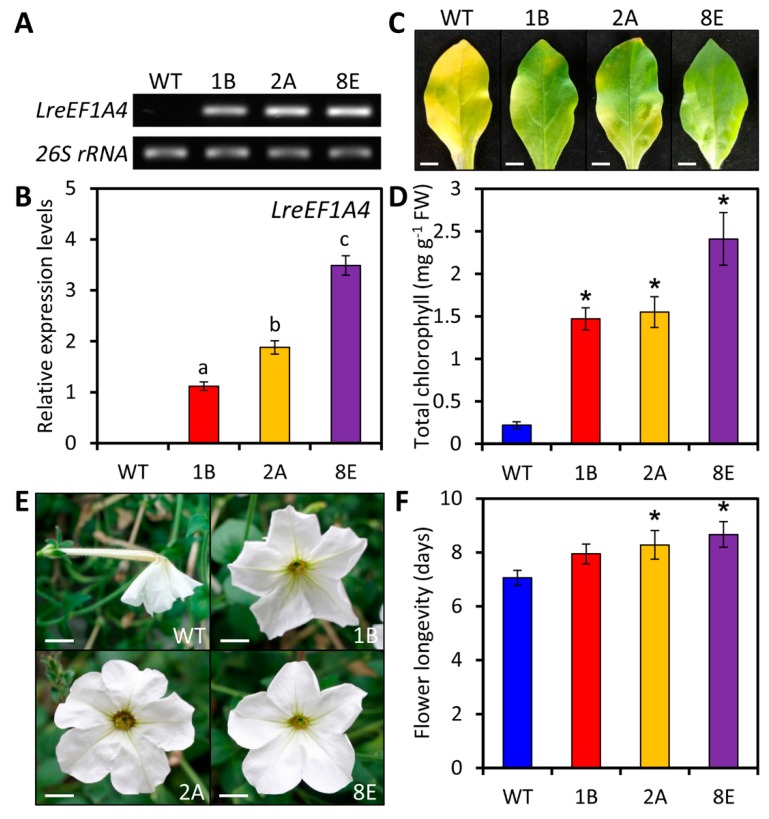
Delayed leaf and flower senescence in transgenic petunia plants overexpressing *LreEF1A4*. Semi-quantitative reverse transcription-polymerase chain reaction (RT-PCR) (**A**) and quantitative real-time PCR (**B**) analyses of *LreEF1A4* expression levels in the leaves of wild-type (WT) and *LreEF1A4*-overexpressing transgenic petunia lines (1B, 2A, and 8E). *26S rRNA* was used as a reference gene. Representative phenotypes (**C**) and total chlorophyll content (**D**) of the leaves at the bottom of 8-week-old WT and transgenic lines. Scale bars = 1.0 cm. Representative phenotypes of attached unpollinated flowers at 8 days after anthesis (**E**) and longevity of flowers (**F**) from WT and transgenic petunia plants. Scale bars = 1.0 cm. Error bars represent standard error (SE) of the mean from three independent biological replicates. Significance of difference was determined using one-way analysis of variance (ANOVA) test (*p* < 0.05) and shown as various letters, or using Student’s *t*-test (*p* < 0.05) and marked by asterisks.

**Figure 5 ijms-21-02083-f005:**
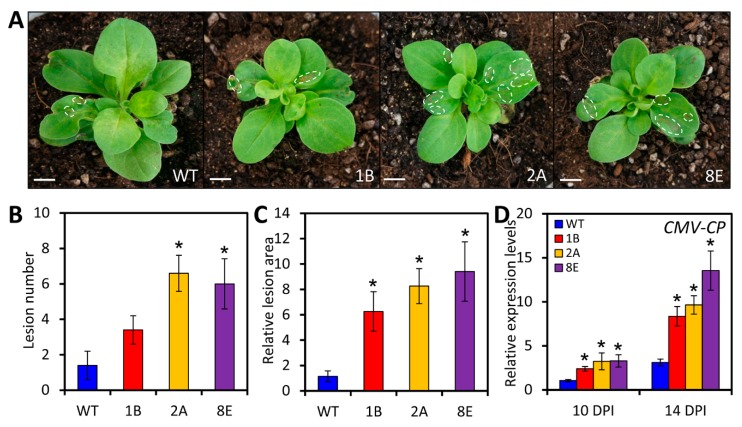
Increased susceptibility to cucumber mosaic virus (CMV) infection in *LreEF1A4*-overexpressing transgenic petunia plants. (**A**) Disease symptoms of wild-type (WT) and transgenic lines (1B, 2A, and 8E) at 14 days post inoculation (DPI) with CMV. The plantlets at the four-leaf stage were used for viral inoculation. The necrotic lesions caused by CMV multiplication are marked in dashed circles. Scale bars = 5.0 mm. Number (**B**) and relative area (**C**) of necrotic lesions in systemically-infected leaves of WT and transgenic lines at 14 DPI. (**D**) Quantitative real-time polymerase chain reaction (PCR) analysis of *CMV*-*CP* transcripts encoding CMV coat protein in uppermost younger leaves of petunia plants at 10 and 14 DPI. *26S rRNA* was used as an internal standard. Error bars represent standard error (SE) of the mean from three independent biological replicates. Asterisks indicate statistical significance as calculated by Student’s *t*-test at *p* < 0.05.

**Figure 6 ijms-21-02083-f006:**
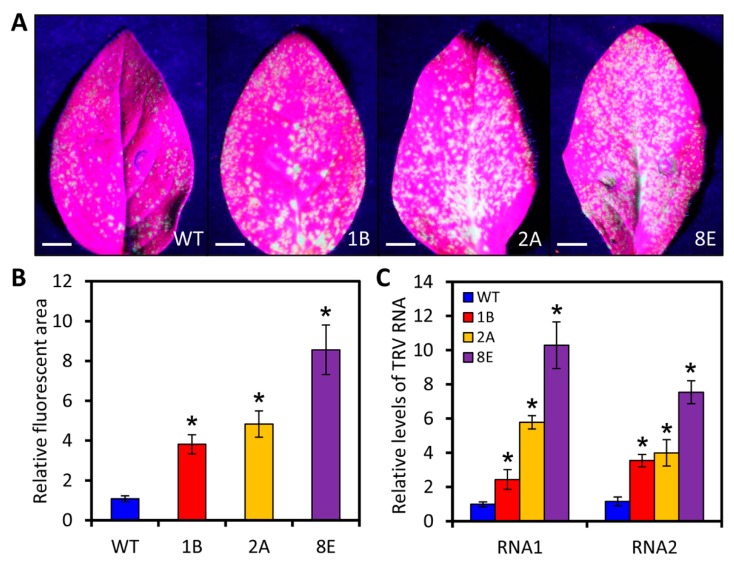
Increased accumulation of tobacco rattle virus-green fluorescent protein (TRV-GFP) in *LreEF1A4*-overexpressing transgenic petunia plants. GFP fluorescence (**A**) and relative fluorescent area (**B**) in TRV-GFP-inoculated leaves of wild-type (WT) and transgenic lines (1B, 2A, and 8E). Photographs were taken at 4 days post inoculation (DPI). Scale bars = 5.0 mm. (**C**) Quantitative real-time polymerase chain reaction (PCR) analysis of TRV RNA1 and RNA2 accumulation levels in the leaves of WT and transgenic petunia plants at 4 DPI. Abundance of *26S rRNA* transcripts was used as an internal control. Error bars represent standard error (SE) of the mean from three independent biological replicates. Significance of difference was evaluated via Student’s *t*-test (*p* < 0.05) and denoted by asterisks.

**Figure 7 ijms-21-02083-f007:**
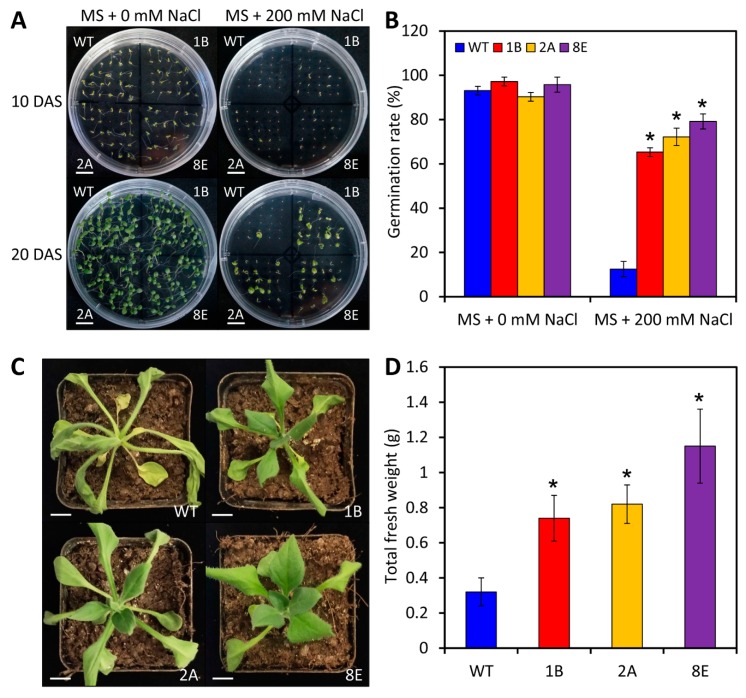
Enhanced tolerance to salt and drought stresses in transgenic petunia plants overexpressing *LreEF1A4*. (**A**) Representative phenotypes of wild-type (WT) and transgenic petunia seedlings (1B, 2A, and 8E) grown in Murashige and Skoog (MS) media containing 0 and 200 mM NaCl. Photographs were taken at 10 and 20 days after sowing (DAS). Scale bars = 1.0 cm. (**B**) Germination percentage of WT and transgenic seedlings at 20 DAS. Representative phenotypes (**C**) and total fresh weight (**D**) of WT and transgenic petunia seedlings at 5 days post treatment with dehydration. Five-week-old plantlets were used for the treatment. Scale bars = 1.0 cm. Error bars represent standard error (SE) of the mean from three independent biological replicates. Asterisks denote statistical significance as determined by Student’s *t*-test (*p* < 0.05).

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
