# Peer review of "LreEF1A4, a Translation Elongation Factor from Lilium regale, Is Pivotal for Cucumber Mosaic Virus and Tobacco Rattle Virus Infections and Tolerance to Salt and Drought"

_ijms, 2020, doi:10.3390/ijms21062083_

Round 1
Reviewer 1 Report
I have reviewed the manuscript "LreEF1A4, a Eukaryotic Translation Elongation Factor from Lilium regale, Is Pivotal for Cucumber Mosaic Virus and Tobacco Rattle Virus Infections and Salt Tolerance". it describes the role of LreEF1A4 as an important regulator for salt resistance and virus proliferation in petunia leaves. the manuscript is very well written with no English errors. Introduction well describes the state of the art of the research area. Results are well described and discussion is complete.
I have only two minor comments:
I was not able to find the the GenBank ID of LreEF1A4, namely "MT083900" (line 14 in the manuscript). The output result for this ID is zero.
Then, I tried to blast the protein sequence to find a correct accession number for LreEF1A4 using the Supplemental data.
In supplemental data, it would be useful to have the fasta of both nucleotide and amino acid sequences. I had to copy the amino acid sequence manually to blast it.
Unfortunately, I was not able to find the sequence corresponding to LreEF1A4.
Is it possible that there is something missing?
Despite these two minor comments, I suggest the editor for publication of the manuscript.
Author Response
Response to Reviewer 1 Comments
Point 1: I was not able to find the the GenBank ID of LreEF1A4, namely "MT083900" (line 14 in the manuscript). The output result for this ID is zero. Then, I tried to blast the protein sequence to find a correct accession number for LreEF1A4 using the Supplemental data.
Response 1: We appreciate the reviewer’s valuable comments. Because the LreEF1A4 sequence was uploaded to the GenBank shortly before submission of our manuscript, it has not been released yet. It’s necessary to explain here that the submitted sequence data to GenBank will be quickly assigned an accession number, which could be then used for any publication. But this submission is not automatically and immediately deposited into GenBank database after being accessioned. Each sequence must be individually examined and processed by the GenBank annotation staff to ensure that it is free of errors or problems. This procedure may take a while before release, and we believe that the sequence data of LreEF1A4 will be found in the GenBank soon.
Point 2: In supplemental data, it would be useful to have the fasta of both nucleotide and amino acid sequences. I had to copy the amino acid sequence manually to blast it. Unfortunately, I was not able to find the sequence corresponding to LreEF1A4. Is it possible that there is something missing?
Response 2: We thank the reviewer for kindly offering such constructive comments, and agreed that the FASTA-formatted nucleotide and amino acid sequences could be convenient for readers to copy and paste. Besides, the LreEF1A4 sequence information is temporarily unavailable at the NCBI website due to the regular submission process of GenBank. To make the readers clearly see and edit the nucleotide and protein sequences of LreEF1A4, we provided them in FASTA format in a new Supplementary Figure S2. The previous Figure S2 and S3 were therefore changed to the current Figure S3 and S4, respectively. The relevant modification was made in the Results, Discussion, and Supplementary Materials sections (Page 3, Line 17; Page 7, Line 4, 6; Page 8, Line 8, 10, 33; Page 10, Line 25; Page 12, Line 16, 22; Page 15, Line 40-42).
Reviewer 2 Report
Sun et al. “LreEF1A4, a eukaryotic translation elongation factor from Lilium regale, is pivotal for Cucumber mosaic virus and Tobacco rattle virus infections and salt tolerance”, International Journal of Molecular Sciences, ijms-738733-v1.
This review paper describes overexpression of the eukaryotic translation elongation factor LreEF1A4 from lily Lilium regale in transgenic petunia Petunia hybrida. Currently, there is a lack of data on the functions of plant translation elongation factors, and, therefore, the topic of this manuscript (Ms) is relevant for International Journal of Molecular Sciences (IJMS). However, there are several important critical remarks to the Ms. I do not recommend this Ms for publication in IJMS in its present form. The Ms needs Major Revision.
- Major Compulsory Revisions
1) In my view, it is necessary to extend the data presented in the Ms text and improve discussion and conclusions. The reason is that the current Ms version needs improvement and extension to correspond to the high rank of the journal of 4.183 impact factor. Please include several additional experiments. For example, please test the transgenic petunia resistance to several abiotic stress factors (not only salt stress). In Discussion, the obtained results should be compared with that in previously published papers (with the available literature data on the effects of overexpression of the eukaryotic translation elongation factors in plants).
2) The authors should include a statistical treatment of the data presented in Fig. 3.
3) Why did the authors used 200 mM NaCl concentration for experiments (Fig. 7)? Please either explain this particular dose or use minimum two concentrations.
4) As I understood (Fig. 3), LreEF1A4 expression significantly increased not only after NaCl treatment but also after cold and drought treatments. Did the authors investigate the resistance of transgenic plants to drought and cold?
5) The Ms contains a number of misprints, grammar mistakes and errors in the writing style. I recommend that the authors should use help of a native English speaker or send the Ms to an English Editing Service to proofread scientific writing.
For example, the title contains mistakes in English and should be improved:
Correct “LreEF1A4, a eukaryotic translation elongation factor from Lilium regale, is pivotal for Cucumber mosaic virus and Tobacco rattle virus infections and salt tolerance” to “LreEF1A4, an eukaryotic translation elongation factor from Lilium regale, is pivotal for Cucumber mosaic virus and Tobacco rattle virus infections and salt tolerance”. I also suggest that the word “eukaryotic” should be removed, since it is clear that Lilium is an eukaryotic organism. Then, “an” should be changed to “a”.
Minor:
6) The authors should improve the legends for figures, e.g. mention used abbreviations (Fig. 3 – “in L. regale leaves under CMV inoculation” correct to “in Lilium regale leaves under Cucumber mosaic virus (CMV) inoculation”.
7) Authors should include distance bars for Fig. 4c, 4e, 6a, 7a.
Author Response
Response to Reviewer 2 Comments
Point 1: In my view, it is necessary to extend the data presented in the Ms text and improve discussion and conclusions. The reason is that the current Ms version needs improvement and extension to correspond to the high rank of the journal of 4.183 impact factor. Please include several additional experiments. For example, please test the transgenic petunia resistance to several abiotic stress factors (not only salt stress). In Discussion, the obtained results should be compared with that in previously published papers (with the available literature data on the effects of overexpression of the eukaryotic translation elongation factors in plants).
Response 1: We thank the reviewer for the valuable comments, and agreed that the present manuscript requires additional experiments for improvement and extension. As suggested by the reviewer, we carried out new experiments to quickly investigate the function of LreEF1A4 in drought and cold tolerance, using the WT and LreEF1A4-overexpressing transgenic petunia seedlings that are currently growing in our growth chamber. The obtained results revealed that LreEF1A4 overexpression resulted in an enhanced tolerance to drought stress in transgenic lines, which exhibited remarkably higher total fresh weight than WT lines following exposure to dehydration (Figure 7C, D). No significant difference in symptoms and electrolyte leakage was observed between WT and transgenic petunia lines under low temperature conditions (Supplementary Figure S5A, B). The new data on drought and cold assays were respectively added to Figure 7 and Supplementary Figure S5. Based on our new findings, the previous manuscript title was accordingly changed to “LreEF1A4, a Translation Elongation Factor from Lilium regale, Is Pivotal for Cucumber Mosaic Virus and Tobacco Rattle Virus Infections and Tolerance to Salt and Drought” (Page 1, Line 2, 4-5). We also made the relevant modification in the Abstract, Keywords, Introduction, Results, Discussion, Materials and Methods, Figure legends, Supplementary Materials, and References sections (Page 1, Line 28-30, 32; Page 3, Line 10; Page 9, Line 11-17, 22-25; Page 10, Line 2, 6-12, 24; Page 11, 35-38, 43-45; Page 12, Line 48-50; Page 13, Line 30-35; Page 15, Line 34, 44; Page 18, Line 50-51; Page 19, Line 26-28). In particular, the obtained results were compared with that in previously published papers in Discussion, and the relevant citations were added in the Discussion and Materials and Methods sections as well (Page 11, Line 36-38; Page 13, Line 34).
Point 2: The authors should include a statistical treatment of the data presented in Fig. 3.
Response 2: We agreed with the reviewer that a statistical analysis would be needed to determine the significance of LreEF1A1-4 expression difference at various time points after CMV infection, abiotic stresses and hormone treatments. So we conducted this analysis using Student’s t-test and the significance was determined at P < 0.05. To clearly show the statistical significance denoted by asterisks, we made new histograms instead of the original line charts in Figure 3A-I. The results of statistical analysis are consistent with the corresponding description in the Results section of our manuscript. We added “Asterisks indicate statistical significance as determined by Student’s t-test at P < 0.05.” at the end of Figure 3 legend (Page 6, Line 16-17).
Point 3: Why did the authors used 200 mM NaCl concentration for experiments (Fig. 7)? Please either explain this particular dose or use minimum two concentrations.
Response 3: We are very appreciative of the reviewer’s insightful question. High salinity, commonly caused by high NaCl concentrations, is one of the major abiotic stresses affecting plant growth and productivity. Numerous researchers have defined the NaCl concentration of 200 mM or more as high salinity level for plants [1-2]. For example, He et al. [3] termed the NaCl solutions of 50 mM, 100 mM, and 200 mM as low, moderate, and high salinity stresses, respectively, to perform the salt treatment in a cereal plant. The concentration of 200 mM NaCl is frequently used for salt assays in crops, including Lilium regale [4] and petunia [5], suggesting that this NaCl concentration could elicit high salinity stress. Therefore, we chose 200 mM NaCl for salt treatment experiments in the present work. Our results of induced LreEF1As expression in L. regale and significant phenotypic difference between WT and transgenic petunia lines upon exposure to salt indicate that the 200 mM NaCl concentration is appropriate.
References:
- Misra, N.; Dwivedi, U.N. Genotypic difference in salinity tolerance of green gram cultivars. Plant Sci. 2004, 166, 1135-1142.
- Gao, Y.; Ma, J.; Zheng, J.C.; Chen, J.; Chen, M.; Zhou, Y.B.; Fu, J.D.; Xu, Z.S.; Ma, Y.Z. The elongation factor GmEF4 is involved in the response to drought and salt tolerance in soybean. Int. J. Mol. Sci. 2019, 20, 3001.
- He, J.F.; Goyal, R.; Laroche, A.; Zhao, M.L.; Lu, Z.X. Effects of salinity stress on starch morphology, composition and thermal properties during grain development in triticale. Can. J. Plant Sci. 2013, 93, 765-771.
- Wei, C.; Cui, Q.; Zhang, X.Q.; Zhao, Y.Q.; Jia, G.X. Three P5CS genes including a novel one from Lilium regale play distinct roles in osmotic, drought and salt stress tolerance. J. Plant Biol. 2016, 59, 456-466.
- Arun, M.; Radhakrishnan, R.; Ai, T.N.; Naing, A.H.; Lee, I.J.; Kim, C.K. Nitrogenous compounds enhance the growth of petunia and reprogram biochemical changes against the adverse effect of salinity. J. Hortic. Sci. Biotechnol. 2016, 91, 562-572.
Point 4: As I understood (Fig. 3), LreEF1A4 expression significantly increased not only after NaCl treatment but also after cold and drought treatments. Did the authors investigate the resistance of transgenic plants to drought and cold?
Response 4: Sure. As we responded to the Point 1 above, we performed new experiments to examine the effect of LreEF1A4 overexpression on the drought and cold stresses, considering the increased transcription of LreEF1A4 following the treatments with these two abiotic factors. The transgenic petunia plants overexpressing LreEF1A4 exhibited promoted tolerance to drought compared to WT. However, the cold tolerance of petunia plants was unaffected by LreEF1A4 overexpression. The new results were added to Figure 7C, D and Supplementary Figure S5A, B, and the corresponding modification was made throughout the whole manuscript (please see Response 1).
Point 5: The Ms contains a number of misprints, grammar mistakes and errors in the writing style. I recommend that the authors should use help of a native English speaker or send the Ms to an English Editing Service to proofread scientific writing. For example, the title contains mistakes in English and should be improved: Correct “LreEF1A4, a eukaryotic translation elongation factor from Lilium regale, is pivotal for Cucumber mosaic virus and Tobacco rattle virus infections and salt tolerance” to “LreEF1A4, an eukaryotic translation elongation factor from Lilium regale, is pivotal for Cucumber mosaic virus and Tobacco rattle virus infections and salt tolerance”. I also suggest that the word “eukaryotic” should be removed, since it is clear that Lilium is an eukaryotic organism. Then, “an” should be changed to “a”.
Response 5: To correct the misprints, grammar mistakes and errors in the original text, we sincerely asked Ayla Norris (University of California-Davis, CA, USA), who is a native English speaker and a researcher in the same field of plant pathology, to help revise the whole manuscript. An extensive editing for the language of this paper has been performed. The descriptions about all the modified parts were provided in the “Other changes made to this manuscript” at the end of this response letter.
As for the title issue of our manuscript, we agreed with the reviewer’s advice and changed the previous “LreEF1A4, a Eukaryotic Translation Elongation Factor from Lilium regale, Is Pivotal for Cucumber Mosaic Virus and Tobacco Rattle Virus Infections and Salt Tolerance” to the current “LreEF1A4, a Translation Elongation Factor from Lilium regale, Is Pivotal for Cucumber Mosaic Virus and Tobacco Rattle Virus Infections and Tolerance to Salt and Drought” (Page 1, Line 2, 4-5).
Point 6: The authors should improve the legends for figures, e.g. mention used abbreviations (Fig. 3 – “in L. regale leaves under CMV inoculation” correct to “in Lilium regale leaves under Cucumber mosaic virus (CMV) inoculation”.
Response 6: According to the review’s kind suggestions, we re-wrote the full names of the words, including “Lilium regale”, “Cucumber mosaic virus”, “standard error”, and “Tobacco rattle virus”, to replace their abbreviations when they firstly appeared in the legends of Figures or Supplementary Figures. (Page 4, Line 3; Page 5, Line 19; Page 6, Line 9; Page 7, Line 15; Page 8, Line 12, 20; Page 9, Line 2, 8-9; Page 10, Line 10; Page 15, Line 43).
Point 7: Authors should include distance bars for Fig. 4c, 4e, 6a, 7a.
Response 7: We added the distance bars in Figure 4C, 4E, 6A, 7A, 7C, and Supplementary Figure S4A, S5A. The relevant statement was provided in the legends of Figures and Supplementary Figures (Page 7, Line 12-14; Page 9, Line 5; Page 10, Line 5, 10).
Other changes made to this manuscript
Page 1, Line 16: We added “the” in front of “pathogenesis”.
Page 1, Line 17: “infections” was added after “Tobacco rattle virus (TRV)”.
Page 1, Line 18-19: “the elongation factor 1 alpha subunit” was changed to “the alpha subunit of elongation factor 1”.
Page 1, Line 19: We added “a” in front of “CMV-elicited suppression subtractive hybridization library”.
Page 1, Line 22-23: “Protein modelling analysis revealed that amino acid substitutions in four LreEF1As may not give rise to significant difference in their enzymatic functions” was modified as “Protein modelling analysis revealed that the amino acid substitutions among four LreEF1As may not affect their enzymatic functions”.
Page 1, Line 25: “transcript levels” was changed to “transcription”.
Page 1, Line 26: “levels” was replaced with “expression”.
Page 1, Line 35: “as” was added after “termed”.
Page 1, Line 37: “broadly existing” was substituted by “widely present”.
Page 1, Line 43: We added a comma after “ribosome”, and modified “to be” as “and is”.
Page 2, Line 5-6: “A direct in vitro interaction of eEF1A with a few proteins, including tubulin [7], calmodulin [8], and actin [9], has been reported” was changed to “Direct in vitro interactions of eEF1A with a few proteins, including tubulin [7], calmodulin [8], and actin [9], have been reported”.
Page 2, Line 18: “shows” and “result in” were changed to “showed” and “resulted in”, respectively.
Page 2, Line 20-21: “many efforts have been made” was revised as “much effort has been made”.
Page 2, Line 25: “demonstrates” was changed to “demonstrated”.
Page 2, Line 29: “the” in front of “eEF1A-virus” was removed.
Page 2, Line 30-31: “the” was added in front of “down-regulation” and “interaction”, respectively.
Page 2, Line 33: “the” in front of “winter barley” was deleted.
Page 2, Line 34-36: We re-stated “The knock-out mutant of AtEF1a in Arabidopsis displays higher sensitivity to NaCl stress, and conversely its overexpressing plants are more tolerant to salt” as “The AtEF1a knock-out mutant of Arabidopsis displays higher sensitivity to NaCl stress, and conversely, the plants overexpressing AtEF1a are more tolerant to salt”.
Page 2, Line 40: “like lily” at the end of the sentence was removed.
Page 2, Line 42-44: “Its economic significance is mainly reflected not only by aesthetic and aromatic values from showy flowers, but also by nutritional and therapeutic benefits from fleshy bulb scales” was changed to “Its economic significance is reflected not only by the aesthetic and aromatic values of the flowers, but also by the nutritional and therapeutic benefits of its fleshy bulb scales”.
Page 2, Line 44: We replaced “Lily comprises” with “The genus Lilium is comprised of”.
Page 2, Line 45: “species” after “55” was removed.
Page 2, Line 50: “the” was added in front of “other nine species”.
Page 2, Line 52: “For the response to” was modified as “Regarding its response to”.
Page 3, Line 10: “accumulation” was changed to “infections”.
Page 3, Line 14: “eukaryotic” was deleted.
Page 4, Line 7: “other three” was adjusted to “three other”.
Page 5, Line 2: “analysis” after “Sequence alignment” was deleted.
Page 5, Line 4: The uppercase word “A” should be lowercase, and was therefore changed to “a”.
Page 5, Line 8-9: “The spatial location of these amino acid sites was examined” was changed to “The spatial locations of these substitution sites were examined”.
Page 5, Line 11-12: “No amino acid substitution site displayed any interacting potential in the substrate-binding (GDP) domain I” was changed to “No amino acid substitution site in the substrate-binding (GDP) domain I displayed any interacting potential with the superimposed substrate”.
Page 5, Line 14: “The results indicate that these amino acid changes” was revised as “These results indicate that the amino acid changes”.
Page 5, Line 27-28: “To compare the difference of LreEF1A4 with other three copies in expression patterns” was modified as “To compare the expression pattern difference of LreEF1A4 with three other copies”.
Page 6, Line 1-2: “A remarkable decrease in four LreEF1As expression” was changed to “A remarkable decrease in the expression of four LreEF1As”.
Page 6, Line 5: “maximum” was changed to “peak”.
Page 6, Line 6-7: We re-stated “No significant alteration was found when measuring the LreEF1As expression” as “No significant alteration of transcription was found for LreEF1As”.
Page 6, Line 19-20: “was performed” was added after “a genetic transformation assay”, and “, was employed” at the end of the sentence was removed.
Page 6, Line 25: “stayed” was changed to “remained”.
Page 7, Line 23-24: “Increased number and size of necrotic lesions were shown” was revised as “An increased number and larger size of necrotic lesions were found”.
Page 7, Line 25-26 and Page 8, Line 1: “This symptom variation was in accordance with the transcript accumulation of CMV-CP encoding CMV coat protein, which was conspicuously increased in LreEF1A4-overexpressing lines at 10 and 14 DPI” was revised as “This symptom variation was in accordance with the measured accumulation of CMV coat protein (CMV-CP) transcripts, which were significantly increased in LreEF1A4-overexpressing lines at 10 and 14 DPI”.
Page 8, Line 4-5: We moved “PheEF1A” up to the back of “down-regulate”, and deleted “, namely PheEF1A”.
Page 8, Line 10: “inoculated with CMV” was changed to “upon CMV inoculation”.
Page 8, Line 25: “could” was changed to “can”.
Page 8, Line 28: “that” was added in front of “in the WT lines”.
Page 8, Line 30: “On the other hand” was changed to “Meanwhile”.
Page 8, Line 32: “in comparison to TRV-EV” was replaced with “compared to the control plants infiltrated with TRV-EV”.
Page 9, Line 7: “Error bars represent SE of the mean from three independent biological replicates.” was redundant and therefore removed.
Page 9, Line 16: “kept” was changed to “maintained”.
Page 9, Line 20-21: “which were drastically lower than the seeds of” was changed to “which was drastically lower than the germination rates of”.
Page 10, Line 14-16: “Large-scale transcriptomics allows for the availability of an enriched population of genes differentially expressed in plants” was revised as “Large-scale transcriptome analysis allows the identification of candidate genes differentially expressed in plants under various stress conditions”.
Page 10, Line 16: “In view of” was substituted by “Considering the”.
Page 10, Line 18-19: “in connection to” was changed to “related to”.
Page 10, Line 21: “the” was added in front of “identification”.
Page 10, Line 24: “transcript” was replaced with “gene”, and “accumulation” was changed to “infection”.
Page 10, Line 26: “comprised of” was changed to “is comprised of”.
Page 11, Line 4-6: We re-stated “But whether these four LreEF1A genes are all members of eEF1A family in L. regale has yet to be defined, due to a gigantic genome size and limited genomic information in Lilium” as “Due to a gigantic genome size and limited genomic information in Lilium, more eEF1A genes may be identified in this species”.
Page 11, Line 7: “eEF1A proteins” was simplified as “eEF1As”.
Page 11, Line 8: “a handful of” was changed to “a number of”, and “in” was replaced with “among”.
Page 11, Line 10-11: “The loop corresponds to” was modified as “The loops correspond to”, and “is” was accordingly changed to “are”.
Page 11, Line 12: We inserted “-“ between “12” and “amino acid”.
Page 11, Line 17: “eEF1A proteins” was changed to “eEF1As”.
Page 11, Line 20: “functional estimation” was changed to “functional assessment”.
Page 11, Line 32: “expression outcomes” was changed to “observations”.
Page 11, Line 51: “Even if in animals” was replaced with “In animals”.
Page 12, Line 4: “virus spread” was changed to “virus propagation”.
Page 12, Line 7: “the” was added in front of “knockdown”.
Page 12, Line 10-11: We replaced “multiplication of a quantity of virus types” with “propagation of multiple virus types”.
Page 13, Line 28: “WT and” was added in front of “T2 transgenic petunia lines”.
Page 14, Line 10: “PDB: 4C0S” was changed to “PDB: 5LZS”.
Page 14, Line 15: The gap between “srv/” and “software” was deleted.
Page 15, Line 28: “In this work” was changed to “In this study”.
Page 15, Line 29: As suggested by the reviewer, “eukaryotic” was removed.
Page 15, Line 35: “the data generated here” was changed to “the data presented here”.
Page 16, Line 7-9: We added “We appreciate Ayla Norris’s kind favor for editing this manuscript. We also thank two anonymous reviewers for the invaluable and helpful comments to improve this manuscript.” in the Acknowledgments section.
Round 2
Reviewer 2 Report
- Minor:
1) In Fig. 3, 4, 5, 6, 7: “Asterisks indicate statistical significance as determined by Student’s t-test at P < 0.05.” – Authors should indicate statistical significance in comparison with what probes.
2) Are there any data on the number of surviving plants after salt and drought stresses? This would significantly enhance the existing results.
Author Response
Response to Reviewer 2 Comments Point 1: In Fig. 3, 4, 5, 6, 7: “Asterisks indicate statistical significance as determined by Student’s t-test at P < 0.05.” – Authors should indicate statistical significance in comparison with what probes. Response 1: We are very grateful to the reviewer for kindly providing the helpful comments. As suggested, we added the relevant statements to clarify what probes were used for comparison of data during statistical significance tests in the legends of Figure 3, 4, 5, 6, 7, and Supplementary Figure S3, S4, S5 (Page 7, Line 1; Page 8, Line 3-5; Page 9, Line 1, 22; Page 11, Line 1-2). Point 2: Are there any data on the number of surviving plants after salt and drought stresses? This would significantly enhance the existing results. Response 2: Thanks for the constructive suggestions. Yes, we obtained the data on the survival rate of WT and LreEF1A4-overexpressing transgenic petunia plants following exposure to stresses. For the salt tolerance test, we measured the germination rate of the plants at 20 days after sowing (DAS) in the NaCl-supplemented medium (Figure 7B). Under normal growth conditions, the petunia seeds begin to germinate at 5-7 DAS in medium. The seeds that had not germinated at 20 DAS could be considered dead to a large extent. Therefore, the germination rate is almost equivalent to the survival rate in our opinion. For the drought tolerance test, we rewatered the WT and transgenic petunia seedlings after the dehydration treatment, and calculated the survival rate of the seedlings 3 days later. We found that transgenic petunia plants overexpressing LreEF1A4 exhibited higher survival rate than WT plants. The new data on survival rate were added to Figure 7E, and the panels of Figure 7 were properly adjusted (Page 10, Line 10). The relevant modification was made in the Results, Figure legend, and Materials and Methods sections (Page 10, Line 6-7, 17-18; Page 13, Line 45-46).